# Instance-Conditional Knowledge Distillation for Object Detection

**Zijian Kang**[*]
Xi'an Jiaotong University
kzj123@stu.xjtu.edu.cn

**Peizhen Zhang**[*]
MEGVII Technology
zhangpeizhen@megvii.com

**Xiangyu Zhang**
MEGVII Technology
zhangxiangyu@megvii.com

**Jian Sun**
MEGVII Technology
sunjian@megvii.com

**Nanning Zheng**
Xi'an Jiaotong University
nnzheng@mail.xjtu.edu.cn

## Abstract

Knowledge distillation has shown great success in classification, however, it is still challenging for detection. In a typical image for detection, representations from different locations may have different contributions to detection targets, making the distillation hard to balance. In this paper, we propose a conditional distillation framework to distill the desired knowledge, namely knowledge that is beneficial in terms of both classification and localization for every instance. The framework introduces a learnable conditional decoding module, which retrieves information given each target instance as query. Specifically, we encode the condition information as query and use the teacher's representations as key. The attention between query and key is used to measure the contribution of different features, guided by a localization-recognition-sensitive auxiliary task. Extensive experiments demonstrate the efficacy of our method: we observe impressive improvements under various settings. Notably, we boost RetinaNet with ResNet-50 backbone from 37.4 to 40.7 mAP ($+3.3$) under $1\times$ schedule, that even surpasses the teacher (40.4 mAP) with ResNet-101 backbone under $3\times$ schedule. Code has been released on https://github.com/megvii-research/ICD.

## 1 Introduction

Deep learning applications blossom in recent years with the breakthrough of Deep Neural Networks (DNNs) [17, 24, 21]. In pursuit of high performance, advanced DNNs usually stack tons of blocks with millions of parameters, which are computation and memory consuming. The heavy design hinders the deployment of many practical downstream applications like object detection in resource-limited devices. Plenty of techniques have been proposed to address this issue, like network pruning [15, 27, 18], quantization [22, 23, 35], mobile architecture design [38, 39] and knowledge distillation (KD) [19, 37, 43]. Among them, KD is one of the most popular choices, since it can boost a target network without introducing extra inference-time burden or modifications.

KD is popularized by Hinton *et al.* [19], where knowledge of a strong pretrained teacher network is transferred to a small target student network in the classification scenario. Many good works emerge following the classification track [50, 37, 32]. However, most methods for classification perform badly in the detection: only slight improvements are observed [28, 51]. This can be attributed to two reasons: (1) Other than category classification, another challenging goal to localize the object is seldomly considered. (2) Multiple target objects are presented in an image for detection, where objects

---

[*]Equal contribution.

35th Conference on Neural Information Processing Systems (NeurIPS 2021).

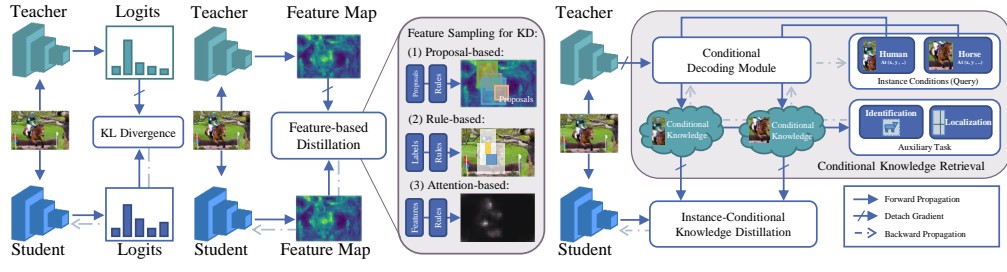

| (a) KD [19]. | (b) Conventional detection KD. | (c) Ours: Instance-conditional KD. |

Figure 1: Compare with different methods for knowledge distillation. (a) KD [19] for classification transfers logits. (b) Recent methods for detection KD distill intermediate features, different region-based sampling methods are proposed. (c) Our method explicitly distill the desired knowledge.

can distribute in different locations. Due to these reasons, the knowledge becomes rather ambiguous and imbalance in detection: representations from different positions like foreground or background, borders or centers, could have different contributions, which makes distillation challenging.

To handle the above challenge, two strategies are usually adopted by previous methods in detection. First, the distillation is usually conducted among intermediate representations, which cover all necessary features for both classification and localization. Second, different feature selection methods are proposed to overcome the imbalance issue. Existent works could be divided into three types according to the feature selection paradigm: proposal-based, rule-based and attention-based. In proposal-based methods [28, 11, 6], proposal regions predicted by the RPN [36] or detector are sampled for distillation. In rule-based methods [14, 45], regions selected by predesigned rules like foreground or label-assigned regions are sampled. Despite their improvements, limitations still exist due to the hand-crafted designs, e.g., many methods neglect the informative context regions or involve meticulous decisions. Recently, Zhang *et al.* [51] propose to use attention [43], a type of intermediate activation of the network, to guide the distillation. Although attention provides inherent hints for discriminative areas, the relation between activation and knowledge for detection is still unclear. To further improve KD quality, we hope to provide an explicit solution to connect the desired knowledge with feature selection.

Towards this goal, we present **I**nstance-**C**onditional knowledge **D**istillation (ICD), which introduces a new KD framework based-on conditional knowledge retrieval. In ICD, we propose to use a decoding network to find and distill knowledge associated with different instances, we deem such knowledge as instance-conditional knowledge. Fig. 1 shows the framework and compares it with former ones, ICD learns to find desired knowledge, which is much more flexible than previous methods, and is more consistent with detection targets. In detail, we design a conditional decoding module to locate knowledge, the correlation between knowledge and each instance is measured by the instance-aware attention via the transformer decoder [5, 43]. In which human observed instances are projected to query and the correlation is measured by scaled-product attention between query and teacher's representations. Following this formulation, the distillation is conducted over features decomposed by the decoder and weighted by the instance-aware attention. Last but not least, to optimize the decoding module, we also introduce an auxiliary task, which teaches the decoder to find useful information for identification and localization. The task defines the goal for knowledge retrieval, it facilitates the decoder instead of the student. Overall, our contribution is summarized in three-fold:

- We introduce a novel framework to locate useful knowledge in detection KD, we formulate the knowledge retrieval explicitly by a decoding network and optimize it via an auxiliary task.

- We adopt the conditional modeling paradigm to facilitate instance-wise knowledge transferring. We encode human observed instances as query and decompose teacher's representations to key and value to locate fine-grained knowledge. To our knowledge, it is the first trial to explore instance-oriented knowledge for detection.

- We perform comprehensive experiments on challenging benchmarks. Results demonstrate impressive improvements over various detectors with up to 4 AP gain in MS-COCO, including recent detectors for instance segmentation [41, 46, 16]. In some cases, students with $1\times$ schedule are even able to outperform their teachers with larger backbones trained $3\times$ longer.

## 2 Related Works

### 2.1 Knowledge Distillation

Knowledge distillation aims to transfer knowledge from a strong teacher to a weaker student network to facilitate supervised learning. The teacher is usually a large pretrained network, who provides smoother supervision and more hints on visual concepts, that improves the training quality and convergence speed [49, 9]. KD for image classification has been studied for years, they are usually categorized into three types [13]: response-based [19], feature-based [37, 43] and relation-based [32].

Among them, feature-based distillation over multi-scale features is adopted from most of detection KD works, to deal with knowledge among multiple instances in different regions. Most of these works can be formulated as region selection for distillation, where foreground-background unbalancing is considered as a key problem in some studies [45, 51, 14]. Under this paradigm, we divide them into three kinds: (1) proposal-based, (2) rule-based and (3) attention-based.

(1) Proposal-based methods rely on the prediction of the RPN or detection network to find foreground regions, *e.g.*, Chen *et al.* [6] and Li *et al.* [28] propose to distilling regions predicted by RPN [36], Dai *et al.* [11] proposes GI scores to locate controversial predictions for distillation.

(2) Rule-based methods rely on designed rules that can be inflexible and hyper-parameters inefficient, *e.g.*, Wang *et al.* [45] distill assigned regions where anchor and ground-truth have a large IoU, Guo *et al.* [14] distill foreground and background regions separately with different factors.

(3) Attention-based methods rely on activations to locate discriminative areas, yet they do not direct to knowledge that the student needs. Only a recent work from Zhang *et al.* [51] considers attention, they build the spatial-channel-wise attention to weigh the distillation.

To overcome the above limitations, we explore the instance-conditional knowledge retrieval formulated by a decoder to explicitly search for useful knowledge. Like other methods, ICD does not have extra cost during inference or use extra data (besides existing labels and a pretrained teacher).

### 2.2 Conditional Computation

Conditional computation is widely adopted to infer contents on a given condition. Our study mostly focuses on how to identify visual contents given an instance as a condition. This is usually formulated as query an instance on the image, *e.g.*, visual question answer [1, 2] and image-text matching [26] queries information and regions specified by natural language. Besides query by language, other types of query are proposed in recent years. For example, DETR [5] queries on fixed proposal embeddings, Chen *et al.* [8] encodes points as queries to facilitate weakly-supervised learning. These works adopt transformer decoder to infer upon global receptive fields that cover all visual contents, yet they usually rely on cascaded decoders that are costly for training. From another perspective, CondInst [41] and SOLOv2 [46] generate queries based on network predictions and achieves great performance on instance segmentation. Different from them, this work adopts the query-based approach to retrieve knowledge and build query based on annotated instances.

### 2.3 Object Detection

Object detection has been developed rapidly. Modern detectors are roughly divided into two-stage or one-stage detectors. Two-stage detectors usually adopt Region Proposal Network (RPN) to generate initial rough predictions and refine them with detection heads, the typical example is Faster R-CNN [36]. On the contrary, one-stage detectors directly predict on the feature map, which are usually faster, they include RetinaNet [30], FCOS [42]. Besides of this rough division, many extensions are introduced in recent years, *e.g.*, extension to instance segmentation [41, 46, 16], anchor-free models [42, 25] and end-to-end detection [5, 44, 20]. Among these works, multi-scale features are usually adopted to enhance performance, *e.g.*, FPN [29], which is considered as a typical case for our study. To generalize to various methods, the proposed method distills the intermediate features and does not rely on detector-specific designs.

# 3 Method

## 3.1 Overview

As discussed in former studies [14, 51], useful knowledge for detection distributes unevenly in intermediate features. To facilitate KD, we propose to transfer instance-conditional knowledge between student and teacher network, termed $\kappa_i^{\mathcal{S}}$ and $\kappa_i^{\mathcal{T}}$ corresponding to the $i_{th}$ instance:

$$\mathcal{L}_{distill} = \sum_{i=1}^{N} \mathcal{L}_d(\kappa_i^{\mathcal{S}}, \kappa_i^{\mathcal{T}}) \tag{1}$$

The knowledge towards teacher's representations $\mathcal{T}$ and condition $y_i$ is formulated as $\kappa_i^{\mathcal{T}} = \mathcal{G}(\mathcal{T}, y_i)$ ($\kappa_i^{\mathcal{S}}$ similarly), where $\mathcal{G}$ is the instance-conditional decoding module, optimized by an auxiliary loss illustrated in Sec. 3.3. The overall framework is shown in Fig. 2.

In the following sections, we will introduce the instance-conditional knowledge (Sec. 3.2), describe the auxiliary task design (Sec. 3.3), and discuss the knowledge transferring (Sec. 3.4).

## 3.2 Instance-conditional Knowledge

In this section, we elaborate the instance-conditional decoding module $\mathcal{G}$, which computes instance-conditional knowledge $\kappa_i^{\mathcal{T}}$ from (1) *unconditional knowledge* given (2) *instance conditions*.

(1) The unconditional knowledge $\mathcal{T}$, symbolizes all available information from the teacher detector. Since modern detectors commonly involve a feature pyramid network (FPN) [29] to extract rich multi-scale representations, we present multi-scale representations as $\mathcal{T} = \{X_p \in \mathbb{R}^{D \times H_p \times W_p}\}_{p \in \mathcal{P}}$, where $\mathcal{P}$ signifies the spatial resolutions while $D$ is the channel dimension. By concatenating representations at different scales along the spatial dimension, we obtain $A^{\mathcal{T}} \in \mathbb{R}^{L \times D}$, where $L = \sum_{p \in \mathcal{P}} H_p W_p$ is the sum of total pixels number across scales.

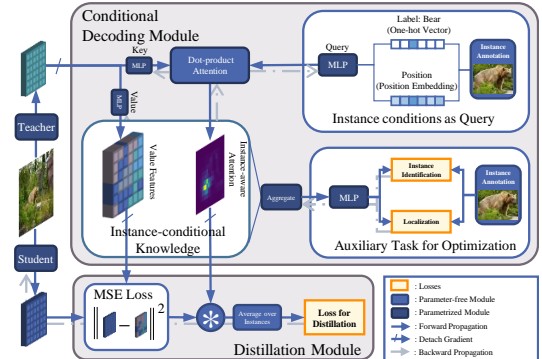

Figure 2: We propose a decoding module to retrieve knowledge via query-based attention, where instance annotations are encoded as a query. An auxiliary task is proposed to optimize the decoding module and the feature-based distillation loss weighted by the attention is used to update student.

(2) The instance condition, originally describing a human-observed object, is denoted by $\mathcal{Y} = \{y_i\}_{i=1}^{N}$, where $N$ is the object number and $y_i = (c_i, \mathbf{b}_i)$ is the annotation for the $i$-th instance, including category $c_i$ and box location $\mathbf{b}_i = (x_i, y_i, w_i, h_i)$ which specifies the localization and size information.

To produce learnable embeddings for each instance, the annotation is mapped to a *query* feature vector $\mathbf{q}_i$ in the hidden space, which specifies a condition to collect desired knowledge:

$$\mathbf{q}_i = \mathcal{F}_q(\mathcal{E}(y_i)), \quad \mathbf{q}_i \in \mathbb{R}^D \tag{2}$$

where $\mathcal{E}(\cdot)$ is a instance encoding function (detailed in in Sec. 3) and $\mathcal{F}_q$ is a Multi-Layer Perception network (MLP).

We retrieve knowledge from $\mathcal{T}$ given $\mathbf{q}_i$ by measuring responses of correlation. This is formulated by dot-product attention [43] with $M$ concurrent heads in a query-key attention manner. In which each head $j$ corresponds to three linear layers ($\mathcal{F}_j^k$, $\mathcal{F}_j^q$, $\mathcal{F}_j^v$) *w.r.t.* the key, query and value construction.

The key feature $K_j^{\mathcal{T}}$ is computed by projecting teacher's representations $A^{\mathcal{T}}$ with the positional embeddings [43, 5] $P \in \mathbb{R}^{L \times d}$ as in Eq. 3, where $\mathcal{F}_{pe}$ denotes a linear projection over position embeddings. The value feature $V_j^{\mathcal{T}}$ and query $\mathbf{q}_{ij}$ is projected by linear mappings to a sub-space with $d = D/M$ channels, $\mathcal{F}_j^v$ on $A^{\mathcal{T}}$ and $\mathcal{F}_j^q$ over $\mathbf{q}_i$ respectively, as shown in Eq. 4. At last, an

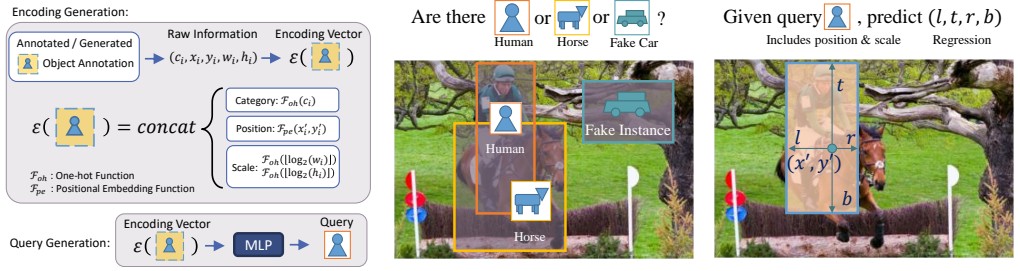

(a) Encoding for instance conditions.      (b) Identification task.      (c) Localization task.

Figure 3: The illustration of the auxiliary task. (a) The instance encoding function encodes a instance condition to a vector, it is then projected as query features. (b) The identification task learns to identify the existence the queried instance. (c) The localization task learns to predict the boundary given an uncertain position provided by the query.

instance-aware attention mask $\mathbf{m}_{ij}$ of the $i$-th instance by the $j$-th head is obtained by normalized dot-product between $\mathrm{K}_j^{\mathcal{T}}$ and $\mathbf{q}_{ij}$:

$$\mathrm{K}_j^{\mathcal{T}} = \mathcal{F}_j^k(\mathrm{A}^{\mathcal{T}} + \mathcal{F}_{pe}(\mathrm{P})), \ \ \mathrm{K}_j^{\mathcal{T}} \in \mathbb{R}^{L \times d} \tag{3}$$

$$\mathrm{V}_j^{\mathcal{T}} = \mathcal{F}_j^v(\mathrm{A}^{\mathcal{T}}), \ \ \mathrm{V}_j^{\mathcal{T}} \in \mathbb{R}^{L \times d} \tag{4}$$

$$\mathbf{q}_{ij} = \mathcal{F}_j^q(\mathbf{q}_i), \ \ \mathbf{q}_{ij} \in \mathbb{R}^d \tag{5}$$

$$\mathbf{m}_{ij} = softmax(\frac{\mathrm{K}_j^{\mathcal{T}}\mathbf{q}_{ij}}{\sqrt{d}}), \ \ \mathbf{m}_{ij} \in \mathbb{R}^L \tag{6}$$

Intuitively, the querying along the key features and value features describes the correlation between representations and instances. We collect $\kappa_i^{\mathcal{T}} = \{(\mathbf{m}_{ij}, \mathrm{V}_j^{\mathcal{T}})\}_{j=1}^M$ as the instance-conditional knowledge from $\mathcal{T}$, which encodes knowledge corresponds to the $i$th instance.

### 3.3 Auxiliary Task

In this section, we introduce the auxiliary task to optimize the decoding module $\mathcal{G}$. First, we aggregate instance-level information to identify and localize objects. This could be obtained by aggregating the instance-conditional knowledge by the function $\mathcal{F}_{agg}$, which includes sum-product aggregation over attention $\mathbf{m}_{ij}$ and $\mathrm{V}_j^{\mathcal{T}}$, concatenate features from each head, add residuals and project with a feed-forward network as proposed in [43, 5]:

$$\mathbf{g}_i^{\mathcal{T}} = \mathcal{F}_{agg}(\kappa_i^{\mathcal{T}}, \mathbf{q}_i), \ \ \mathbf{g}_i^{\mathcal{T}} \in \mathbb{R}^D \tag{7}$$

To let the instance-level aggregated information $\mathbf{g}_i^{\mathcal{T}}$ retain sufficient instance cues, one could design an instance-sensitive task to optimize it as below:

$$\mathcal{L}_{aux} = \mathcal{L}_{ins}(\mathbf{g}_i^{\mathcal{T}}, \mathcal{H}(\mathbf{y}_i)) \tag{8}$$

where $\mathcal{H}$ encodes the instance information as targets. However, directly adopt Eq. 8 might lead to trivial solution, since $\mathbf{y}_i$ is accessible from both $\mathbf{g}_i^{\mathcal{T}}$ (through $\mathbf{q}_i$, see Eq. 2) and $\mathcal{H}(\mathbf{y}_i)$. It is possible that parameters will learn a shortcut, that ignore the teacher representations $\mathcal{T}$. To resolve this issue, we propose to drop information on encoding function $\mathcal{E}(\cdot)$, to force the aggregation function $\mathcal{F}_{agg}$ to excavate hints from $\mathcal{T}$.

The information dropping is adopted by replacing the accurate annotation for instance conditions to uncertain ones. For bounding box annotations, we relieve them to rough box centers with rough scales indicators. The rough box center $(x_i', y_i')$ is obtained by random jittering as shown below:

$$\begin{cases} x_i' = x_i + \phi_x w_i, \\ y_i' = y_i + \phi_y h_i, \end{cases} \tag{9}$$

where $(w_i, h_i)$ is the width and height of the bounding box and $\phi_x, \phi_y$ are sampled from a uniform distribution $\Phi \sim U[-a, a]$, where we set a=0.3 empirically. The scales indicators are obtained by

rounding the box sizes in the logarithmic space, *i.e.*, $\lfloor log_2(w_i) \rfloor$, $\lfloor log_2(h_i) \rfloor$. In addition, to let the decoder learn to identify instances and be aware of the uncertainty, we generate fake instances for the identification task according to dataset distributions, this is detailed in Appendix A. As a result, it collect coarse information and obtain instance encoding through $\mathcal{E}(\cdot)$ as depicted in Fig. 3a. where $c_i$ is the category, $\mathcal{F}_{oh}$ is the one hot vectorization, *concat* is the concatenation operator and $\mathcal{F}_{pe}$ is the position embedding function.

Finally, the aggregated representation $\mathbf{g}_i^{\mathcal{T}}$ are optimized by the auxiliary task. We introduces two predictors denoted by $\mathcal{F}_{obj}$ and $\mathcal{F}_{reg}$ respectively to predict identification and localization results. We adopt binary cross entropy loss (BCE) to optimize the real-fake identification and $L1$ loss to optimize the regression.

$$\mathcal{L}_{aux} = \mathcal{L}_{BCE}(\mathcal{F}_{obj}(\mathbf{g}_i^{\mathcal{T}}), \delta_{obj}(\mathbf{y}_i)) + \mathcal{L}_1(\mathcal{F}_{reg}(\mathbf{g}_i^{\mathcal{T}}), \delta_{reg}(\mathbf{y}_i)) \tag{10}$$

where $\delta_{obj}(\cdot)$ is an indicator, it yields 1 if $\mathbf{y}_i$ is real and 0 otherwise. Following common practice, the localization loss for fake examples is ignored. $\delta_{reg}$ is the preparing function for regression targets, following [42]. Appendix A provides more implementation details.

### 3.4 Instance-Conditional Distillation

Lastly, we present the formulation for conditional knowledge distillation. We obtain the projected value features $V_j^{\mathcal{S}} \in \mathbb{R}^{L \times d}$ of the student representations analogous to Eq. 4 in Sec. 3.2. By adopting the instance-aware attention mask as a measurement of correlations between feature and each instance, we formulate the distillation loss as value features mimicking guided by the attention:

$$\mathcal{L}_{distill} = \frac{1}{MN_r} \sum_{j=1}^{M} \sum_{i=1}^{N} \delta_{obj}(\mathbf{y}_i) \cdot < \mathbf{m}_{ij}, \mathcal{L}_{MSE}(V_j^{\mathcal{S}}, V_j^{\mathcal{T}}) > \tag{11}$$

where $N_r = \sum_{i=1}^{N} \delta_{obj}(\mathbf{y}_i)$, $(N_r \leq N)$ is the real instances number, $\mathcal{L}_{MSE}(V_j^{\mathcal{S}}, V_j^{\mathcal{T}}) \in \mathbb{R}^L$ is the pixel-wise mean-square error along the hidden dimension[2] and $< \cdot, \cdot >$ is the Dirac notation for inner product. For stability, the learnable variable $\mathbf{m}_{ij}$ and $V_j^{\mathcal{T}}$ are detached during distillation. Combine with the supervised learning loss $\mathcal{L}_{det}$, the overall loss with a coefficient $\lambda$ is summarized below:

$$\mathcal{L}_{total} = \mathcal{L}_{det} + \mathcal{L}_{aux} + \lambda \mathcal{L}_{distill} \tag{12}$$

It is worth noticing, only the gradients *w.r.t.* $\mathcal{L}_{distill}$ and $\mathcal{L}_{det}$ back-propagate to the student network (from representations $\mathcal{S}$). The gradients of $\mathcal{L}_{aux}$ only update the instance-conditional decoding function $\mathcal{G}$ and auxiliary task related modules.

## 4 Experiments

### 4.1 Experiment Settings

We conduct experiments on Pytorch [34] with the widely used Detectron2 library [47] and AdelaiDet library [3] [40]. All experiments are running on eight 2080ti GPUs with 2 images in each. We adopt the $1\times$ scheduler, which denotes 9k iterations of training, following the standard protocols in Detectron2 unless otherwise specified. Scale jittering with random flip is adopted as data augmentation.

For distillation, the hyper-parameter $\lambda$ is set to 8 for one-stage detectors and 3 for two-stage detectors respectively. To optimize the transformer decoder, we adopt AdamW optimizer [33] for the decoder and MLPs following common settings for transformer [43, 5]. Corresponding hyper-parameters follows [5], where the initial learning rate and weight decay are set to 1e-4. We adopt the 256 hidden dimension for our decoder and all MLPs, the decoder has 8 heads in parallel. The projection layer $\mathcal{F}_q$ is a 3 layer MLP, $F_{reg}$ and $F_{obj}$ share another 3 layer MLP. In addition, we notice some newly initialized modules of the student share the same size of the teacher, *e.g.*, the detection head, FPN.

---

[2]$\mathcal{L}_{MSE}(.)$ is conducted over normalized features with parameter-free LayerNorm [3] for stabilization.

[3]All libraries are open-sourced and public available, please refer to citations for more details.

Table 1: Comparison with previous methods on challenging benchmark MS-COCO. The method proposed by Li *et al.* [28] does not apply to RetinaNet. † denotes the inheriting strategy.

| Method | Faster R-CNN [36] | | | | RetinaNet [30] | | | |
|---|---|---|---|---|---|---|---|---|
| | AP | $AP_S$ | $AP_M$ | $AP_L$ | AP | $AP_S$ | $AP_M$ | $AP_L$ |
| Teacher w. ResNet-101 (3×) | 42.0 | 25.2 | 45.6 | 54.6 | 40.4 | 24.0 | 44.3 | 52.2 |
| Student w. ResNet-50 (1×) | 37.9 | 22.4 | 41.1 | 49.1 | 37.4 | 23.1 | 41.6 | 48.3 |
| + FitNet [37] | 39.3 (+1.4) | 22.7 | 42.3 | 51.7 | 38.2 (+0.8) | 21.8 | 42.6 | 48.8 |
| + Li *et al.* [28] | 39.5 (+1.5) | 23.3 | 43.0 | 51.4 | - | - | - | - |
| + Wang *et al.* [45] | 39.2 (+1.3) | 23.2 | 42.8 | 50.4 | 38.4 (+1.0) | 23.3 | 42.6 | 49.1 |
| + Zhang *et al.* [51] | 40.0 (+2.1) | 23.2 | 43.3 | 52.5 | 39.3 (+1.9) | 23.4 | 43.6 | 50.6 |
| + Ours | 40.4 (+2.5) | 23.4 | 44.0 | 52.0 | 39.9 (+2.5) | 25.0 | 43.9 | 51.0 |
| + Ours † | 40.9 (+3.0) | 24.5 | 44.2 | 53.5 | 40.7 (+3.3) | 24.2 | 45.0 | 52.7 |

Table 2: Experiments on more detectors with ICD. Type denotes the AP score is evaluated on bounding box (`BBox`) or instance mask (`Mask`). † denotes using the inheriting strategy.

| Detector | Setting | Type | AP | $AP_{50}$ | $AP_{75}$ | $AP_S$ | $AP_M$ | $AP_L$ |
|---|---|---|---|---|---|---|---|---|
| FCOS [42]
Teacher: 18.8 FPS / 51.2M
Student: 25.0 FPS / 32.2M | Teacher (3×) | BBox | 42.6 | 61.6 | 45.8 | 26.2 | 46.3 | 53.8 |
| | Student (1×) | | 39.4 | 58.2 | 42.4 | 24.2 | 43.4 | 49.4 |
| | + Ours | | 41.7(+2.3) | 60.3 | 45.4 | 26.9 | 45.9 | 52.6 |
| | + Ours † | | 42.9(+3.5) | 61.6 | 46.6 | 27.8 | 46.8 | 54.6 |
| Mask R-CNN [16]
Teacher: 17.5 FPS / 63.3M
Student: 22.9 FPS / 44.3M | Teacher (3×) | BBox | 42.9 | 63.3 | 46.8 | 26.4 | 46.6 | 56.1 |
| | Student (1×) | | 38.6 | 59.5 | 42.1 | 22.5 | 42.0 | 49.9 |
| | + Ours | | 40.4 (+1.8) | 60.9 | 44.2 | 24.4 | 43.7 | 52.0 |
| | + Ours † | | 41.2 (+2.6) | 62.0 | 45.0 | 25.1 | 44.5 | 53.6 |
| | Teacher (3×) | Mask | 38.6 | 60.4 | 41.3 | 19.5 | 41.3 | 55.3 |
| | Student (1×) | | 35.2 | 56.3 | 37.5 | 17.2 | 37.7 | 50.3 |
| | + Ours | | 36.7 (+1.5) | 58.0 | 39.2 | 18.4 | 38.9 | 52.5 |
| | + Ours † | | 37.4 (+2.2) | 58.7 | 40.1 | 19.1 | 39.8 | 53.7 |
| SOLOv2 [46]
Teacher: 16.6 FPS / 65.5M
Student: 21.4 FPS / 46.5M | Teacher (3×) | Mask | 39.0 | 59.4 | 41.9 | 16.2 | 43.1 | 58.2 |
| | Student | | 34.6 | 54.7 | 36.9 | 13.2 | 37.9 | 53.3 |
| | + Ours | | 37.2 (+2.6) | 57.6 | 39.8 | 14.8 | 40.7 | 57.0 |
| | + Ours † | | 38.5 (+3.9) | 59.0 | 41.2 | 15.9 | 42.3 | 58.9 |
| CondInst [41]
Teacher: 16.8 FPS / 53.5M
Student: 21.3 FPS / 34.1M | Teacher (3×) | BBox | 44.6 | 63.7 | 48.4 | 27.5 | 47.8 | 58.4 |
| | Student (1×) | | 39.7 | 58.8 | 43.1 | 23.9 | 43.3 | 50.1 |
| | + Ours | | 42.4 (+2.7) | 61.5 | 46.1 | 25.3 | 46.0 | 54.3 |
| | + Ours † | | 43.7 (+4.0) | 62.9 | 47.2 | 27.1 | 47.3 | 56.6 |
| | Teacher (3×) | Mask | 39.8 | 61.4 | 42.6 | 19.4 | 43.5 | 58.3 |
| | Student (1×) | | 35.7 | 56.7 | 37.7 | 16.8 | 39.1 | 50.3 |
| | + Ours | | 37.8 (+2.1) | 59.1 | 40.4 | 17.5 | 41.4 | 54.7 |
| | + Ours † | | 39.1 (+3.4) | 60.5 | 42.0 | 19.1 | 42.6 | 57.0 |

We find initialize these modules with teacher's parameters will lead to faster convergence, we call it the inheriting strategy in experiments.

Most experiments are conducted on a large scale object detection benchmark **MS-COCO** [4][31] with 80 classes. We train models on MS-COCO 2017 `trainval115k` subset and validate on `minival` subset. Following common protocols, we report mean Average Precision (AP) as an evaluation metric, together with AP under different thresholds and scales, *i.e.*, $AP_{50}$, $AP_{75}$, $AP_S$, $AP_M$, $AP_L$. Other experiments are listed in Appendix B, e.g., on VOC [12] and Cityscapes [10], more ablations.

### 4.2 Main Results

**Compare with state-of-the-art methods.** We compare our method (ICD) with previous state-of-the-arts (SOTAs), including a classic distillation method FitNet [37], two typical detection KD methods [28, 45], and a very recent work with strong performance from Zhang *et al.* [51]. The comparison is conducted on two classic detectors: Faster R-CNN [36] and RetinaNet [30]. We adopt

---

[4]MS-COCO is publicly available, the annotations are licensed under a `Creative Commons Attribution 4.0 License` and the use of the images follows `Flickr Terms of Use`. Refer to [31] for more details.

Table 3: Experiments on mobile backbones. † denotes using the inheriting strategy.

| Detector | Setting | Backbone | AP | $AP_{50}$ | $AP_{75}$ | $AP_S$ | $AP_M$ | $AP_L$ |
|---|---|---|---|---|---|---|---|---|
| RetinaNet [30] | Teacher (3×) | ResNet-101 [17] | 40.4 | 60.3 | 43.2 | 24.0 | 44.3 | 52.2 |
| | Student (1×) | | 26.4 | 42.0 | 27.8 | 13.8 | 28.8 | 34.1 |
| | + Ours | MBV2 [38] | 29.5 (+3.1) | 45.5 | 31.2 | 16.2 | 32.2 | 38.3 |
| | + Ours † | | 31.6 (+5.2) | 48.5 | 33.4 | 17.6 | 34.7 | 41.3 |
| RetinaNet [30] | Teacher (3×) | ResNet-101 [17] | 40.4 | 60.3 | 43.2 | 24.0 | 44.3 | 52.2 |
| | Student (1×) | | 34.9 | 54.8 | 37.0 | 20.9 | 38.9 | 44.8 |
| | + Ours | Eff-B0 [39] | 36.7 (+1.8) | 56.0 | 38.7 | 21.1 | 40.6 | 48.1 |
| | + Ours † | | 38.0 (+3.1) | 57.5 | 40.2 | 22.4 | 41.6 | 50.3 |
| FRCNN [36] | Teacher (3×) | ResNet-101 [17] | 42.0 | 62.5 | 45.9 | 25.2 | 45.6 | 54.6 |
| | Student (1×) | | 27.2 | 44.7 | 28.8 | 14.6 | 29.6 | 35.6 |
| | + Ours | MBV2 [38] | 30.2 (+3.0) | 48.0 | 32.5 | 17.0 | 32.2 | 39.1 |
| | + Ours † | | 31.4 (+4.2) | 49.4 | 33.6 | 17.6 | 33.5 | 41.3 |
| FRCNN [36] | Teacher (3×) | ResNet-101 [17] | 42.0 | 62.5 | 45.9 | 25.2 | 45.6 | 54.6 |
| | Student (1×) | | 35.3 | 56.8 | 37.8 | 20.8 | 38.2 | 45.1 |
| | + Ours | Eff-B0 [39] | 37.0 (+1.7) | 58.0 | 39.6 | 21.1 | 40.0 | 48.3 |
| | + Ours † | | 37.9 (+2.6) | 58.7 | 40.8 | 21.4 | 40.9 | 49.5 |

detectron2 official released models with ResNet-101 backbone trained on 3× scheduler as teacher networks, the student is trained on 1× with ResNet-50 backbone following the above settings.

As shown in Table 1, ICD brings about 2.5 AP and 3.0 AP improvement for plain training and training with the inheriting strategy respectively. Especially for RetinaNet, the student with distillation even outperforms a strong teacher trained on 3× scheduler. Compare with previous SOTAs, the proposed method leads to a considerable margin for about 0.5 AP without the inheriting strategy.

**Results on other settings.** We further evaluate ICD under various detectors, *e.g.*, a commonly used anchor-free detector FCOS [42], and three detectors that have been extended to instance segmentation: Mask R-CNN [16], SOLOv2 [46] and CondInst [41]. We adopt networks with ResNet-101 on 3× scheduler as teachers and networks with ResNet-50 on 1× scheduler as students following the above settings. As shown in Table 2, we observe consistent improvements for both detection and instance segmentation. There are at most around 4 AP improvement on CondInst [41] on object detection and SOLOv2 [46] on instance segmentation. Moreover, students with weaker backbone (ResNet-50 *v.s.* ResNet-101) and less training images (1/3) even outperform (FCOS) or perform on par (SOLOv2, CondInst) with teachers. Note that ICD does not introduce extra burden during inference, our method improves about 25% of FPS[5] and reduces 40% of parameters compared with teachers.

**Mobile backbones.** Aside from main experiments on commonly used ResNet [17], we also conduct experiments on mobile backbones, which are frequently used in low-power devices. We evaluate our method on two prevalent architectures: MobileNet V2 (MBV2) [38] and EfficientNet-B0 (Eff-B0) [39]. The latter one adopts the MobileNet V2 as basis, and further extends it with advanced designs like stronger data augmentation and better activations.

Experiments are conducted on Faster R-CNN (*abbr.*, FRCNN) [36] and RetinaNet [30] following the above settings. As shown in Table 3, our method also significantly improves the performance on smaller backbones. For instance, we improve the RetinaNet with MobileNet V2 backbone with 5.2 AP gain and 3.1 AP gain with and without inheriting strategy respectively, and up to 3.1 AP gain for EfficientNet-B0. We also observe consistent improvements over Faster R-CNN, with up to 4.2 AP gain for MobileNet-V2 and 2.6 AP gain for EfficientNet-B0.

## 4.3 Ablation Studies

To verify the design options and the effectiveness of each component, we conduct ablation studies with the classic RetinaNet detector on MS-COCO following the above settings.

**Design of the auxiliary task.** To better understand the role of our auxiliary task, we conduct experiments to evaluate the contribution of each sub-task. Specifically, our auxiliary task is composed

---

[5]FPS is evaluated on Nvidia Tesla V100.

of an identification task with binary cross-entropy loss and a localization task with regression loss, the localization task is further augmented with a hint on bounding box scales. As shown in Table 4, the identification task itself leads to 2.2 AP gain compare with the baseline, this high-light the importance of knowledge on object perception. The regression task itself leads to 1.8 AP gain, and the scale information boosts it for extra 0.2 AP gain. Combine two of them, we achieve 2.5 AP overall improvement, which indicates the fusion of two knowledge brings extra benefits. Note the auxiliary task only update the decoder and does not introduce extra data, which is very different from multitask learning, *e.g.*, Mask R-CNN[16].

Table 4: Comparison with different auxiliary task designs.

| Identification | Localization | + Scale | AP | $AP_{50}$ | $AP_{75}$ | $AP_S$ | $AP_M$ | $AP_L$ |
|---|---|---|---|---|---|---|---|---|
| | | | 37.4 | 56.7 | 40.3 | 23.1 | 41.6 | 48.3 |
| ✓ | | | 39.6 | 59.2 | 42.8 | 23.4 | 44.0 | 50.4 |
| | ✓ | | 39.2 | 58.6 | 42.4 | 23.1 | 43.5 | 50.3 |
| | ✓ | ✓ | 39.4 | 58.8 | 42.4 | 23.2 | 43.8 | 50.5 |
| ✓ | ✓ | ✓ | 39.9 | 59.4 | 43.1 | 25.0 | 43.9 | 51.0 |

**Impact of the instance-aware attention.** To verify the effectiveness of instance-aware attention learned by conditional computation, we directly replace it with different variations: the fine-grained mask in [45], pixel-wise attention activation [51, 43], foreground mask analog to [14] and no attention mask. The result in Fig. 5 shows our instance-aware mask leads to about 0.9 AP gain over the baseline and 0.4 AP gain compare with the best replacement.

Table 5: Comparison with different types of attention.

| Attention Type | AP | $AP_{50}$ | $AP_{75}$ | $AP_S$ | $AP_M$ | $AP_L$ |
|---|---|---|---|---|---|---|
| No Attention | 39.0 | 58.4 | 42.1 | 23.5 | 43.2 | 49.9 |
| Foreground Mask | 39.4 | 58.9 | 42.4 | 23.5 | 43.6 | 50.0 |
| Fine-grained Mask [45] | 39.5 | 59.0 | 42.4 | 23.4 | 43.8 | 50.2 |
| Attention Activation [43] | 39.3 | 58.6 | 42.3 | 22.6 | 43.6 | 50.1 |
| Instance-conditional Attention | 39.9 | 59.4 | 43.1 | 25.0 | 43.9 | 51.0 |

**Design of the decoder.** We mainly verify two properties of the decoder design, as shown in Table 6. First, we find a proper number of heads is important for the best performance. The head number balances the dimension for each sub-space and the number of spaces. The best number lies around 8, which is consistent with former studies [5]. Second, we evaluate the effectiveness of cascaded decoders, as it is widely adopted in former studies [43, 5, 8]. However, we do not find significant differences between different cascade levels, it might because we use a fixed teacher to provide knowledge, that limited the learning ability.

Table 6: Ablation on the number of heads and cascade levels.

(a) Number of heads.

| Heads | AP | $AP_S$ | $AP_M$ | $AP_L$ |
|---|---|---|---|---|
| 1 | 39.4 | 23.7 | 43.8 | 50.5 |
| 4 | 39.7 | 24.0 | 43.9 | 51.2 |
| 8 | 39.9 | 25.0 | 43.9 | 51.0 |
| 16 | 39.7 | 23.6 | 44.2 | 50.5 |

(b) Number of cascade levels.

| Levels | AP | $AP_S$ | $AP_M$ | $AP_L$ |
|---|---|---|---|---|
| 1 | 39.9 | 25.0 | 43.9 | 51.0 |
| 2 | 39.9 | 24.1 | 44.0 | 51.4 |
| 4 | 39.8 | 24.3 | 44.0 | 50.9 |

## 5 Discussion

**Qualitative analysis.** We present a visualization of our learned instance-aware attention in Fig. 4. We find different heads attend to different components related to each instance, *e.g.*, salient parts, boundaries. This might suggest that salient parts are important and knowledge for regression mostly lies around boundaries, instead of averaged on the foreground as assumed in former studies. The head (vi) and (vii) are smoother around key regions, which may relate to context. The head (iv) mainly attends on image corners, which might be a degenerated case or relate to some implicit descriptors.

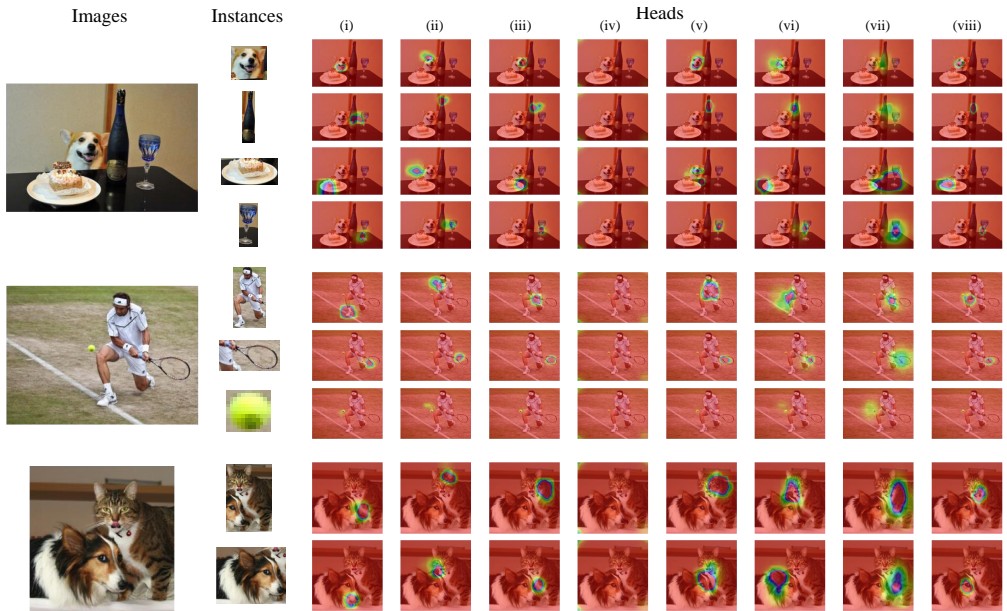

Figure 4: Visualization of our learned instance-aware attention over each head. Red denotes weak areas and pink denotes strong areas.

**Resource consumption.** We benchmark the training speed for our distillation method. As shown in Fig. 5, training the decoder introduces negligible cost. Specifically, we benchmark on $1\times$ scheduler on RetinaNet [30] with eight 2080ti, following the same configuration in Section 4.2. The major time consumption is spent on training the teacher ($3\times$), which takes about 33 hours. Training (forward and backward) the student takes about 8 hours, while training decoder and distillation only take 1.3 hours. Besides the training time, memory consumption of our method is also limited. One can update the decoder part and the student part in consecutive iterations, leaving only intermediate features for distillation in memory across iterations.

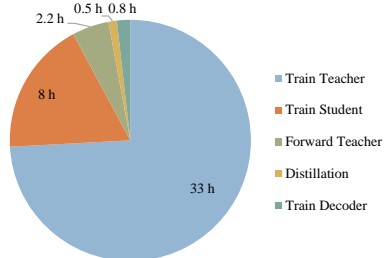

Figure 5: Time consumption.

**Real-world impact.** The proposed method provides a convenient approach to enhance a detector under a certain setting, resulting in that a small model can perform on par with a larger one. On the positive side, it allows users to replace a big model with a smaller one, which reduces energy consumption. On the potentially negative side, the teacher could be costly in training, also the student might inherit biases from the teacher, which is hard for tracing.

## 6 Conclusion

We introduce a novel framework for knowledge distillation. The proposed **I**nstance-**C**onditional knowledge **D**istillation (ICD) method utilizes instance-feature cross attention to select and locate knowledge that correlates with human observed instances, which provides a new framework for KD. Our formulation encodes instance as query and teacher's representation as key. To teach the decoder how to find knowledge, we design an auxiliary task that relies on knowledge for recognition and localization. The proposed method consistently improves various detectors, leading to impressive performance gain, some student networks even surpass their teachers. In addition to our design of the auxiliary task, we believe there are other alternations that can cover different situations or provide a theoretical formulation of knowledge, which will be a potential direction for further researches.

## Acknowledgments and Disclosure of Funding

This paper is supported by the National Key R&D Plan of the Ministry of Science and Technology (Project No. 2020AAA0104400) and Beijing Academy of Artificial Intelligence (BAAI).

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
