# A  Appendix: Details for Auxiliary Task

In this section, we present more details of the auxiliary task. As noted in the paper, the auxiliary task is optimized by an identification loss and a regression loss, which is formulates as:

$$\mathcal{L}_{aux} = \mathcal{L}_{idf} + \mathcal{L}_{reg} \tag{13}$$

$$= \mathcal{L}_{BCE}(\mathcal{F}_{obj}(\mathbf{g}_i^{\mathcal{T}}), \delta_{obj}(\mathbf{y}_i)) + \mathcal{L}_1(\mathcal{F}_{reg}(\mathbf{g}_i^{\mathcal{T}}), \delta_{reg}(\mathbf{y}_i)) \tag{14}$$

where $\mathcal{L}_{idf}$ is the identification loss and $\mathcal{L}_{reg}$ is the regression loss.

**Identification loss.**    The identification task aims to the identify the objectiveness of each instance. The target $\delta_{obj}(\mathbf{y}_i))$ encodes the binary classification labels, denoting whether the instance is randomly generated or manually annotated:

$$\delta_{obj}(\mathbf{y}_i) = \left\{ \begin{array}{ll} 0, & \text{if } \mathbf{y}_i \text{ is randomly sampled.} \\ 1, & \text{if } \mathbf{y}_i \text{ is human annotated.} \end{array} \right. \tag{15}$$

We simply use the binary cross entropy loss averaged over each instance as the identification loss:

$$\mathcal{L}_{idf} = -\frac{1}{N}\sum_{i=1}^{N}\delta_{obj}(\mathbf{y}_i)\log\left(\mathcal{F}_{obj}(\mathbf{g}_i^{\mathcal{T}})\right) + (1 - \delta_{obj}(\mathbf{y}_i))\log\left(1 - \mathcal{F}_{obj}(\mathbf{g}_i^{\mathcal{T}})\right) \tag{16}$$

where $\mathcal{F}_{obj}(\cdot)$ is a prediction function with sigmoid activation.

**Regression loss.**    The regression task aims to locate object boundaries. The design of the regression loss is analogous to [5, 42, 8]. We adopt relative distance as distance metrics following [5, 8], all coordinates are normalized by the image size, *e.g.*, horizontal coordinates along the x-axis are normalized by the image width, rendering all valid coordinates in range $[0, 1]$.

First, each bounding box is presented as a 4D coordinate $(x_i^1, y_i^1, x_i^2, y_i^2)$ for instance $\mathbf{y}_i$, which is composed of the left-top corner position $(x_i^1, y_i^1)$ and the right-bottom corner position $(x_i^2, y_i^2)$. Next, Given the disturbed box center $(x_i', y_i')$, the regression target can be written as position offsets relative to the box center. Finally, we introduce a MLP network $\mathcal{F}_{reg}(\cdot)$ with sigmoid activation to predict these offsets:

$$\delta_{reg}(\mathbf{y}_i) = [l_i, t_i, r_i, b_i] = [x_i' - x_i^1, y_i' - y_i^1, x_i^2 - x_i', y_i^2 - y_i'] \in \mathbb{R}^4 \tag{17}$$

$$\mathcal{F}_{reg}(\mathbf{g}_i^{\mathcal{T}}) = [l_i', t_i', r_i', b_i'] \in \mathbb{R}^4 \tag{18}$$

Following the above formulation, the localization loss with L1 distance is written as:

$$\mathcal{L}_{reg} = \frac{1}{N_r}\sum_{i=1}^{N}\delta_{obj}(\mathbf{y}_i)\left(\frac{|l_i - l_i'|}{w_i} + \frac{|t_i - t_i'|}{h_i} + \frac{|r_i - r_i'|}{w_i} + \frac{|b_i - b_i'|}{h_i}\right) \tag{19}$$

where $w_i$ and $h_i$ is the width and height of each bounding box, $N_r = \sum_{i=1}^{N}\delta_{obj}(\mathbf{y}_i)$ is the number of human annotated instances. Note that we only optimize the regression loss for real instances.

**Fake instances.**    We present more details on how to sample fake instances for the instance identification task. The instance $\mathbf{y}_i$ is either drawn from human annotations, or randomly sampled. We empirically sample five fake instances for every single real instance. In details, the category of each fake instance is drawn from a distribution, where the probability for each class is proportional to their frequency in the training set. At the same time, the location for each fake instance is generated by sampling a center position and a bounding box size. Empirically, we assume that widths and heights of bounding boxes for each class obey the Gaussian distribution. We calculate the statistical mean and variance according to the training set and draw the fake bounding box sizes from the distribution. Besides, center positions are sampled inside the image uniformly. Lastly, we clip bounding boxes that grow outside the image.

**Warm-up for distillation.** We train our decoding modules with the auxiliary task jointly with the detector to reduce computation, yet decoding modules take some time to converge, which may cause unstable training at initial iterations. To avoid it, the distillation loss is introduced after the first 1000 iterations of training. Moreover, we introduce an extra warm-up stage to initialize the decoding modules for smaller datasets like Pascal VOC [12] and Cityscapes [10], to ensure the convergence of decoding modules. In the warm-up stage, we only train decoding modules with the auxiliary task loss, which is much faster than regular training (only takes 26% of the time for each iteration). Teacher and student are fixed in the warm-up stage for the fair comparison.

# B  Appendix: Extended Experiments

**More datasets.** We report results on smaller datasets to verify the effectiveness on different environments, we adopt Pascal VOC [12] to benchmark performance on common scenes and Cityscapes [10] to benchmark performance on street views. Pascal VOC[6] is a classic detection benchmark that includes 20 classes. We use the union of VOC 2012 and VOC 2007 `trainval` for training, VOC 2007 `test` for evaluation. We evaluate our method on Cityscapes[7] to benchmark performance on the outdoor environment, performance is evaluated on instance segmentation. All experimental settings follow Detectron2 [47], networks are trained for 18k iterations on VOC and 24k iterations on Cityscapes. We introduce an extra 6k iterations for auxiliary task warm-up for smaller datasets to stabilize training.

We conduct experiments and comparison on VOC with two widely used detectors: Faster R-CNN and RetinaNet. As shown in Table 7, students with smaller backbones surpass corresponding teachers on both detectors. Combined with the inheriting strategy, the gap further expands. Our method also outperforms SOTAs in VOC, which proves the method's efficacy and robustness.

Table 7: Comparison with other methods on VOC. † denotes using inheriting strategy.

| Method | Backbone | Faster R-CNN [36] | | | RetinaNet [30] | | |
|---|---|---|---|---|---|---|---|
| | | AP | $AP_{50}$ | $AP_{75}$ | AP | $AP_{50}$ | $AP_{75}$ |
| Teacher | ResNet-101 | 56.3 | 82.7 | 62.6 | 57.1 | 82.2 | 63.0 |
| Student | ResNet-50 | 54.2 | 82.1 | 59.9 | 54.7 | 81.1 | 59.4 |
| FitNet [37] | ResNet-50 | 55.0 | 82.2 | 61.2 | 56.4 | 81.7 | 61.7 |
| Li *et al.* [28] | ResNet-50 | 56.2 | 82.9 | 61.8 | - | - | - |
| Wang *et al.* [45] | ResNet-50 | 55.3 | 82.1 | 61.1 | 55.6 | 81.4 | 60.5 |
| Zhang *et al.* [51] | ResNet-50 | 55.4 | 82.0 | 61.3 | 56.7 | 81.9 | 61.9 |
| Ours | ResNet-50 | 56.4 | 82.2 | 63.4 | 57.7 | 82.4 | 63.5 |
| Ours† | ResNet-50 | 57.3 | 82.9 | 63.4 | 58.5 | 82.7 | 64.5 |

We evaluate our method on Cityscapes with Mask R-CNN [16] and CondInst [41] for instance segmentation on street views. Experiments are conducted on MobileNet V2 to fit low-power devices. Besides, due to the limitation of the dataset scale, the trade-off for stronger models is unattractive (only 0.3 AP gap between ResNet-50 and ResNet-101 on Mask R-CNN). As shown in table 8, our method consistently improves over the baseline, with up to 3.6 AP gain on Mask R-CNN. The gap between teacher and student for CondInst [41] is smaller, yet our method also improves it for 1.6 AP,

Table 8: Experiments on Cityscapes with our method. † denotes using the inheriting strategy.

| Method | Backbone | Mask R-CNN [16] | | CondInst [30] | |
|---|---|---|---|---|---|
| | | AP | $AP_{50}$ | AP | $AP_{50}$ |
| Teacher | ResNet-50 | 33.5 | 60.8 | 33.7 | 60.0 |
| Student | MobileNet V2 | 28.8 | 55.2 | 30.5 | 55.5 |
| Ours | MobileNet V2 | 32.2 | 60.1 | 31.1 | 57.1 |
| Ours † | MobileNet V2 | 32.4 | 60.1 | 32.1 | 57.9 |

**FPN-free architectures.** In above experiments, we examine most prevailing architectures for distillation, most of them are equiped with FPN [29] to imporve performace. To verify the generalization

---

[6]Images and annotations are publicly available, the use of images follows the `Flickr Terms of Use`.
[7]Cityscapes is public available for academic purposes, refer to [10] for more details.

ability of ICD on FPN-free architectures, we conduct experiments on two Faster R-CNN variations denoted by C4 and DC5. For Faster R-CNN C4, the fourth resblock of the backbone is embedded in the RoI head, RPN, RoI head and distillation module is stacked at the third block. For Faster R-CNN DC5, the last ResNet block is dilated with stride 2, RPN, RoI head and distillation module is stacked at the last block. As shown in Table 9, our method significantly improves the baseline for 2.8 and 2.0 AP seperately for C4 and DC5.

Table 9: Experiments with different teacher-student pairs on MS-COCO. We do not use the inheriting strategy for fair illustration.

| Teacher | | | Student | | | Distillation | |
|---|---|---|---|---|---|---|---|
| Detector | Backbone | AP | Detector | Backbone | AP | AP | $\Delta$AP |
| Faster R-CNN (3×) | ResNet-101 | 42.0 | Faster R-CNN (1×) | ResNet-50 | 37.9 | 40.4 | +2.5 |
| Cascade Mask R-CNN (3×) | ResNet-101 | 45.5 | Faster R-CNN (1×) | ResNet-50 | 37.9 | 40.6 | +2.7 |
| RetinaNet (3×) | ResNet-101 | 40.4 | RetinaNet (1×) | ResNet-50 | 37.4 | 39.9 | +2.5 |
| FCOS (3×) | ResNet-101 | 42.6 | RetinaNet (1×) | ResNet-50 | 37.4 | 39.7 | +2.3 |

**Distillation across architectures.** ICD is designed to transfer knowledge between detectors with similar detection heads, the situation naturally facilities the feature alignment. However, it is also possible to apply ICD across different detectors. As shown in Table 10, ICD also achieves considerable performance gain despite detection heads are different.

Table 10: Experiments with FPN-free variations on MS-COCO. † denotes the inheriting strategy.

| Method | Backbone | Sche. | Faster R-CNN C4 [36] | | | | Faster R-CNN DC5 [36] | | | |
|---|---|---|---|---|---|---|---|---|---|---|
| | | | AP | $AP_S$ | $AP_M$ | $AP_L$ | AP | $AP_S$ | $AP_M$ | $AP_L$ |
| Teacher | ResNet-101 | 3× | 41.1 | 22.2 | 45.5 | 55.9 | 40.6 | 22.9 | 45.1 | 54.1 |
| Student | ResNet-50 | 1× | 35.7 | 19.2 | 40.9 | 48.7 | 37.3 | 20.1 | 41.7 | 50.0 |
| Ours | ResNet-50 | 1× | 37.1 | 20.9 | 41.9 | 50.7 | 38.8 | 20.9 | 43.3 | 52.5 |
| Ours † | ResNet-50 | 1× | 38.5 | 20.6 | 43.1 | 53.0 | 39.3 | 20.9 | 44.1 | 53.5 |

**Longer schedules.** We further examine our method on longer training periods, as shown in Table 11, the improvement obtained by our method is still impressive. Specifically, the student with RetinaNet detector outperforms its teacher by 0.6 AP and the student with Faster R-CNN performs on par with the teacher. The gain of the inheriting strategy is impressive on 1× scheduler, yet it becomes smaller under sufficient training iterations (3×), which shows the improvement on accelerating training.

Table 11: Experiments with different schedulers on MS-COCO. † denotes the inheriting strategy.

| Method | Backbone | Sche. | Faster R-CNN [36] | | | | RetinaNet [30] | | | |
|---|---|---|---|---|---|---|---|---|---|---|
| | | | AP | $AP_S$ | $AP_M$ | $AP_L$ | AP | $AP_S$ | $AP_M$ | $AP_L$ |
| Teacher | ResNet-101 | 3× | 42.0 | 25.2 | 45.6 | 54.6 | 40.4 | 24.0 | 44.3 | 52.2 |
| Student | ResNet-50 | 1× | 37.9 | 22.4 | 41.1 | 49.1 | 37.4 | 23.1 | 41.6 | 48.3 |
| Student | ResNet-50 | 3× | 40.2 | 24.2 | 43.5 | 52.0 | 38.7 | 23.3 | 42.3 | 50.3 |
| Ours | ResNet-50 | 1× | 40.4 | 23.4 | 44.0 | 52.0 | 39.9 | 25.0 | 43.9 | 51.0 |
| Ours † | ResNet-50 | 1× | 40.9 | 24.5 | 44.2 | 53.5 | 40.7 | 24.2 | 45.0 | 52.7 |
| Ours | ResNet-50 | 3× | 41.8 | 24.9 | 45.4 | 54.4 | 40.8 | 24.3 | 44.6 | 52.8 |
| Ours † | ResNet-50 | 3× | 41.8 | 24.7 | 45.1 | 54.6 | 41.0 | 25.1 | 44.7 | 53.0 |

**Sensitivity of hyperparameters.** ICD introduces a hyperparameter $\lambda$ to control the distillation strength. We conduct a sensitivity analyze of this hyperparameter on MS-COCO with Faster R-CNN and RetinaNet in Table 12. The result shows the performance is robust across a wide range of $\lambda$.

## C   Appendix: Overview of recent methods.

For a fair comparison of recent methods in Sec. 4, we re-implement recent methods on the same baseline with same teacher-student pairs and training settings. As for complement, we also list their

Table 12: Sensitivity analyze of hyper-parameter $\lambda$ on MS-COCO. * denotes adopted settings.

| Method | Backbone | Faster R-CNN [36] | | | | | RetinaNet [30] | | | | |
| --- | --- | --- | --- | --- | --- | --- | --- | --- | --- | --- | --- |
| | | $\lambda$ | AP | $AP_S$ | $AP_M$ | $AP_L$ | $\lambda$ | AP | $AP_S$ | $AP_M$ | $AP_L$ |
| Teacher | ResNet-101 (3×) | - | 42.0 | 25.2 | 45.6 | 54.6 | - | 40.4 | 24.0 | 44.3 | 52.2 |
| Student | ResNet-50 (1×) | - | 37.9 | 22.4 | 41.1 | 49.1 | - | 37.4 | 23.1 | 41.6 | 48.3 |
| ICD | ResNet-50 (1×) | 1 | 40.5 | 24.1 | 43.9 | 53.1 | 2 | 40.5 | 24.0 | 44.8 | 52.2 |
| ICD* | ResNet-50 (1×) | 3 | 40.9 | 24.5 | 44.2 | 53.5 | 6 | 40.7 | 24.2 | 45.0 | 52.7 |
| ICD | ResNet-50 (1×) | 5 | 40.7 | 23.5 | 43.8 | 53.8 | 12 | 40.6 | 23.9 | 44.8 | 52.5 |

settings and paper-reported performance in Table 13, which shows the difference in settings and baselines. Compare with former settings, ICD adopts the 1× scheduler with multi-scale jitters (ms), all training settings follow the standard protocols and we use public released models as our teachers. The baseline has similar performance compared with others and our method outperforms most of the methods with much less training time (2× *v.s.* 1×). We also conduct experiments on similar 2× single-scale (ss) training settings and adopt strong teachers (as suggested by [51]) for Faster R-CNN[8], ICD still shows impressive performance with strong final AP and gains. In addition, we highlight that there are many design variations as suggested by former studies, *e.g.*, [51, 11, 6] adopt multiple loss functions in parallel for distillation, Guo *et al.*[14] adopt complicated rules for foreground-background balancing. The complicated design usually achieves higher performance, yet they also introduce extra hyperparameters and detector-dependant designs that are hard for tuning, while our method only uses a simple distillation loss and only introduces one $\lambda$ to control the distillation strength.

Table 13: Overview of recent methods on MS-COCO. We summarize choices of teacher-student pairs, training schedulers, library and number of hyperparameters (Num.) of recent methods. RCNN denotes Faster R-CNN [36], MRCNN denotes Mask R-CNN [16] and CMRCNN denotes Cascade Mask R-CNN [4]. 2×* denotes the 1× scheduler with 32 batch size.

| Method | Teacher | | | | Student | | | | Distillation | | | |
| --- | --- | --- | --- | --- | --- | --- | --- | --- | --- | --- | --- | --- |
| | Detector | Backbone | Available | AP | Detector | Backbone | Scheduler | AP | Library | Num. | AP | ΔAP |
| Chen *et al.*(2017) [6] | FRCNN | VGG16 | No | 24.2 | FRCNN | VGGM | - | 16.1 | - | 3+ | 17.3 | +1.2 |
| Wang *et al.*(2019) [45] | FRCNN | R101 | No | 34.4 | FRCNN | R101-h | - | 28.8 | Yang *et al.*[48] | 1+ | 31.6 | +2.8 |
| Zhang *et al.*(2021) [51] | CMRCNN | X101-dcn | Yes | 47.3 | FRCNN | R50 | 2× SS | 38.4 | mmdet [7] | 3+ | 41.5 | +3.1 |
| Dai *et al.*(2021) [11] | FRCNN | R101 | No | 39.6 | FRCNN | R50 | 2× SS | 38.3 | - | 3+ | 40.2 | +1.9 |
| Guo *et al.*(2021) [14] | FRCNN | R152 | No | 41.3 | FRCNN | R50 | 2×* SS | 37.4 | mmdet [7] | 5+ | 40.9 | +3.5 |
| Ours | FRCNN | R101 | Yes | 42.0 | FRCNN | R50 | 1× MS | 37.9 | Detectron2 [47] | 1+ | 40.9 | +3.0 |
| Ours | MRCNN | R101 | Yes | 48.9 | FRCNN | R50 | 2× SS | 38.2 | Detectron2 [47] | 1+ | 42.2 | +4.0 |
| Zhang *et al.*(2021) [51] | RetinaNet | X101-dcn | Yes | 41.0 | RetinaNet | R50 | 2× SS | 37.4 | mmdet [7] | 3+ | 39.6 | +2.2 |
| Dai *et al.*(2021) [11] | RetinaNet | R101 | No | 38.1 | RetinaNet | R50 | 2× SS | 36.2 | - | 3+ | 39.1 | +2.9 |
| Guo *et al.*(2021) [14] | RetinaNet | R152 | No | 40.5 | RetinaNet | R50 | 2×* SS | 37.4 | mmdet [7] | 5+ | 39.7 | +3.2 |
| Ours | RetinaNet | R101 | Yes | 40.4 | RetinaNet | R50 | 1× MS | 37.4 | Detectron2 [47] | 1+ | 40.7 | +3.3 |
| Ours | RetinaNet | R101 | Yes | 40.4 | RetinaNet | R50 | 2× SS | 36.2 | Detectron2 [47] | 1+ | 40.5 | +4.3 |

---

[8]We use released models from Detectron2 new baselines as stronger teachers.