# OpenReview forum: "Instance-Conditional Knowledge Distillation for Object Detection"
_NeurIPS.cc/2021/Conference — NeurIPS 2021 Poster_

### Official Review · Reviewer_WeoR · 2021-07-12

**Rating:** 6
**Confidence:** 4

**Summary:**

This paper proposes a new method named Instance-Conditional knowledge Distillation (ICD) to deal with the knowledge distillation problem in image object detection task. Motivated by the fact the traditional classification-level distillation methods cannot focus on the locations of objects, they insert additional instance information to the distillation training together with auxiliary tasks, i.e. recognition and localization. Their proposed distillation framework improves the performance of the student network, and even outperforms the teacher network under some specific settings.

**Ethics Review Area:**

["Discrimination / Bias / Fairness Concerns"]

**Limitations And Societal Impact:**

As the authors mention in Section 5, one limitation of their method is that the student might inherit biases from the teacher which could be hard to identify. Another limitation I can imagine is the access to instance information. What if we cannot access the training label? Where can we get the instance information?

**Main Review:**

**Originality**: This method is original in this area, although it combines several well-known techniques. Related works are also well-summarized and discussed.

**Quality**: With my experience, this method is technically correct. The motivation of is clear and straightforward. Their experiment supports their statements in their methodology. There are two things that might improve the quality of this paper:

(1)	The analysis of why the student network outperforms the teacher network. Is it possible that the teacher network is not well-trained, and the new modules and tasks added in the KD framework make up for the previous training deficiencies?

(2)	The analysis of the hyper-parameter $\lambda$ in equation (12).

**Clarity**: In general, the paper is well-organized and easy to follow. However, I have some suggestions about the figures:

(1)	In Figure 1, the last column uses ”object-oriented“ as its title, but this term does not appear in the introduction part. I only find ”task-oriented“ described in the context. I am not sure if the last column represents their proposed method.

(2)	In Figure 1, there is no description of the blue and gray blocks. I assume the blue one is the student network and the gray one is the teacher network.

(3)	In Figure 2, there are a lot of different kinds of arrows and rectangles. Without any descriptions, it is really hard to understand the details of the proposed method.

**Significance**: The significance and novelty are the main reason for my current rating. Although their experiment indeed shows improvement compared with the 4 previous methods, the contribution of the proposed method is still limited. If I understand the method correctly, their improvement mainly comes from the auxiliary tasks of instance information, which is not a fundamental modification.


**Time Spent Reviewing:**

3

---

> ### Author Response · Authors · 2021-08-10
> **Response to reviewer WeoR. Thanks for the feedback!**
>
> We thank the reviewer for the careful review and valuable comments. We appreciate that the reviewer finds our method original, our motivation is straightforward and clear, and our experiment supports our statements. We hope our answer can address the reviewer’s concerns.
>
> ------------
> **Recap:**
> As a short recap, our main contribution is the attention-based knowledge retrieval for KD, which is based on instance conditions as query. It is considered novel by most reviewers. The adopted modules for knowledge retrieval introduce additional parameters, therefore we propose an auxiliary task to optimize them. Notably, the auxiliary task does not directly provide knowledge to the student.
>
> ---------------
>
> **Q1: The improvement mainly comes from the auxiliary tasks of instance information.**
>
> A: We notice that KD is also considered as an “auxiliary task/loss” sometimes, for clarity, we shall emphasize that _the improvement of the student comes from the KD loss (eq. 11)_ instead of the auxiliary loss (eq. 8), as the latter one does not backpropagate to the student (noted in line no. 206).
>
> We also emphasize that:
>
> 1) our method does not rely on extra labels or data;
>
> 2) the decoding modules exist only during training as a conventional KD framework does;
>
> 3) knowledge is retrieved from a pretrained teacher;
>
> For the above reasons, our method can be cataloged as a novel KD method, very different from typical multi-task training (e.g. Mask R-CNN [1]). It outperforms previous studies under fair comparisons.
>
>
> **Q2: Limitation of the training label.**
>
> A: We follow the supervised training protocol as former KD studies, labels are always available as they are mandatory for the supervised training. Notably, we do not introduce extra labels or data.
>
> Yet we agree that the mentioned limitation might exist in other areas, like in semi-supervised learning, where pseudo labels might be preferred. It is an interesting topic, but it is beyond our consideration.
>
>
> **Q3: Where can we get the instance information?**
>
> A: The instance information is encoded by training labels (annotations), each annotated object is described by a category and a location, detailed in line no. 177-185.
>
>
> **Q4: Why does the student outperform the teacher, is the teacher well-trained?**
>
> **Is the teacher well-trained?**
>
> A: Our teacher networks are well-trained. We adopt networks trained under the standard protocols on long 3x schedules as our teacher, the performance of these networks matches the papers. Moreover, most of our teachers are officially provided by Detectron2 [2] or AdelaiDet [3], which are publicly available and have been adopted in many studies.
>
> **Is it possible that the teacher network is not well-trained, and the new modules and tasks make up for the previous training deficiencies?**
>
> A: First, our teacher networks have been well trained. The training protocols for our teachers are the same as our students', following the standard settings. Second, it is widely known that KD is a training technique that improves supervised training. Our method is further improved from the previous KD methods, the proposed method is more effective in knowledge transferring as shown in Table 1, it improves various SOTA detectors as shown in Table 2, and our ablation in Sec. 4.3 identifies the effectiveness of the proposed modules.
>
> **Why does the student outperform the teacher?**
>
> A: We attribute the strong improvement on our students to three reasons:
>
> 1) According to [4], KD serves as a form of label smoothing. Our method might behave like a smoother detection loss that assists instance-wise supervision. Note that students in [4] also outperform their teachers.
>
> 2) KD assists visual concept learning and provides a better optimization direction, as noted in [5]. Our method improves the concept quality and optimization by adding instance-conditional attention during KD.
>
> 4) Distilling a strong teacher to a small student might serve as an implicit regularization, that enhances the parameter robustness.
>
> Thanks for the suggestion, we will discuss it in the paper.
>
>
> **Q5: Analysis of the hyper-parameter $\lambda$.**
>
> A: Our method only has one hyper-parameters $\lambda$ that controls the distillation strength, we choose the hyper-parameter via grid search. Our method is robust across different values of  $\lambda$,  experiments on MS-COCO are shown below.
>
> | Lambda  | Faster R-CNN (AP)  | Lambda  | RetinaNet(AP)  |
> |---------|--------------------|---------|----------------|
> | 1       | 40.5               | 2       | 40.5           |
> | 3*      | 40.9               | 8*      | 40.7           |
> | 5       | 40.7               | 12      | 40.6           |
>
> ( * is the adopted setting )
>
> Thanks for the suggestion, we will discuss it in the manuscript.
>
> ------
> ### Figures:
>
> **Q6: Blue and gray blocks in Figure 1. Is the blue one is the student network and the gray one is the teacher network?**
>
> A: Actually, the blue block at the top denotes the teacher, while the gray one denotes the student.
>
> Specifically, bottom-up yellow arrows denote that the student is mimicking the teacher, this is usually done via a fitting layer following FitNet [6].
>
> Thanks for the suggestion, we will revise our figure, add labels and descriptions, to make it easier to understand.
>
>
> **Q7: Is the last column with ‘Object oriented’ in Figure 1 denotes our method?**
>
> A: Yes, the last column denotes our method. The ‘object oriented’ denotes that the proposed KD is conditioned by annotated objects, which aims to transfer instance-related knowledge.
>
> Thanks for the suggestion, we will add more descriptions and revise our phrase.
>
>
> **Q8: Arrows in Figure 2.**
>
> A: All the arrows denote the computation flow. Thanks for the suggestion, we will revise the style of all arrows and add more descriptions.
>
> ------------------
>
> ### References:
>
> [1] He et al. Mask r-cnn. ICCV 2017.
>
> [2] Wu et al. Detectron2. 2019.
>
> [3] Tian et al. AdelaiDet: A Toolbox for Instance-level Recognition Tasks. 2019.
>
> [4] Yuan et al. Revisiting knowledge distillation via label smoothing regularization. CVPR 2020.
>
> [5] Cheng et al. Explaining knowledge distillation by quantifying the knowledge. CVPR 2020.
>
> [6] Romero et al. Fitnets: Hints for thin deep nets. ICLR 2015.

---

> > ### Comment · Reviewer_WeoR · 2021-08-13
> > **Raised my rating. Thank the authors for replying.**
> >
> > I appreciate the authors for providing such a detailed response to my comments, especially giving some background knowledge about the KD area. I decide to raise my score to 6 after considering the response from the authors and the comments from other reviewers.
> > * (1) For the novelty of this paper, I am convinced that the proposed instance-conditional KD is a new method. Due to my limited background information, I may not fully understand the core contribution of this paper before. The response from the authors indeed helps a lot.
> > * (2) Thank the authors for adding the analysis of the hyper-parameter $\lambda$. It seems the value of $\lambda$ can vary from 1 to 12 without causing much influence. So is it really necessary?
> > * (3) Thank the authors for addressing my comments on the figures. I believe a better figure helps the readers quickly catch the main idea and contribution.

---

> > > ### Author Response · Authors · 2021-08-14
> > > **Response to reviewer WeoR. Thanks for the reply.**
> > >
> > > Thanks for the reviewer's reply, we appreciate the reviewer's acknowledgment. We will keep on answering questions and revising our manuscript.
> > >
> > > ---------
> > >
> > > **Q: It seems the value of  $\lambda$ can vary from 1 to 12 without causing much influence. So is it really necessary?**
> > >
> > > A: We believe it is reasonable to keep $\lambda$ for potential improvement. First, although our method is robust, tuning $\lambda$ results in better performance. Second, searching for only one hyper-parameter does not cost too many resources. Third, $\lambda$ could be an important factor to generalize on different detectors.
> > >
> > > ----------------
> > >
> > > We hope our reply can address the reviewer's concern.

---

### Official Review · Reviewer_LAyY · 2021-07-16

**Rating:** 6
**Confidence:** 5

**Summary:**

This submission proposes a novel knowledge distillation approach for object detection. The authors attempt to use an auxiliary task with the key-query (transformer-liked) method to determine where and how the feature map can be transferred from a teacher to a student. In the paper, the target instances are encoded as queries to determine the knowledge distilled from the teacher. Extensive experiments show the effectiveness of the proposed approach.

**Limitations And Societal Impact:**

The proposed distillation method only explores the situation where teacher and student have the same framework but different backbones, such that from Resnet101-based teachers to ResNet50-based students. The inheriting parameters strategy is not possible to use if the teacher and student have different detection heads and FPNs.

**Main Review:**

Strength:

+ Novel idea. The authors propose to make use of the transformer-liked method to generate the attention mask based on real instances such that the approach can locate the informative knowledge to guide the student network's training.
+ Promising results and convincing ablation study.

Weakness:

- The English writing of this paper should be improved. Several typos in the paper, such as 'wildly' in line 99, 'Direc' in line 201. Please check the paper carefully.
- Figures 1, 2, and 3 are hard to follow. Please add related notations along with the text and some sub captions to refine them.
- I believe that generating fake instances is crucial in the method, how and why 5? It would be better to discuss more on this part.
- The inheriting parameters strategy, which improves the performance significantly, can only be used in the scenario where the student and teacher share the same architecture in detection head and FPN. It is more or less a big limitation in the knowledge distillation domain where the student and teacher may vary a lot in the network architecture.

**Time Spent Reviewing:**

5

---

> ### Author Response · Authors · 2021-08-10
> **Response to reviewer LAyY. Thanks for the feedback!**
>
> We thank the reviewer for the careful review and valuable comments. We appreciate that the reviewer finds our idea novel, our results promising and our ablation convincing.  We hope our answer can address the reviewer’s concerns.
>
>  ------------------
>
> **Q1: Discussion of fake instances, how and why it is important?**
>
> A: We generate fake instances for three reasons:
>
> 1) Identifying instances is important for detection, many regions in the image are ambiguous and foreground/background imbalance is a known issue. The loss based on real-fake identification encourages the module to find confident regions and decompose features related to potential instances.
>
> 2) The teacher is imperfect, which sometimes makes mistakes. If we only use real instances, the attention can not identify the wrong predictions of the teacher.
>
> 3) Encoding category information as a condition improves the query quality. Classification-based losses will easily lead to trivial solutions if the category has been encoded in query, yet the identification-based is still robust.
>
> We can see in Table 3, the identification loss itself produces decent performance gain. We will discuss it in the manuscript, thanks for the suggestion.
>
>
> **Q2: Limitation of the inheriting strategy.**
>
> A: Yes, the inheriting strategy is limited to the standard condition only, where the teacher and the student share a similar architecture in detection head and FPN.
>
> Yet, our method still outperforms previous methods without the strategy. It is also possible to apply our method across different detectors, experiments on MS-COCO are shown below.
>
> | Student Arch.  | Faster R-CNN   | Faster R-CNN        | RetinaNet    | RetinaNet    |
> |----------------|----------------|---------------------|--------------|--------------|
> | Teacher Arch.  | Faster R-CNN   | Cascade Mask R-CNN  | RetinaNet    | FCOS         |
> | Teacher AP     | 42.0           | 45.5                | 40.4         | 42.6         |
> | Student AP     | 37.9           | 37.9                | 37.4         | 37.4         |
> | + ours AP      | 40.4 (+2.5)    | 40.6 (+2.7)         | 39.9 (+2.5)  | 39.7 (+2.3)  |
>
> We also agree that knowledge transferring among very different networks is an interesting topic for future researches.
>
>
> **Q3: Writing.**
>
> A: Thanks for the suggestion and correction, we will revise the manuscript.
>
>
> **Q4: Figure.**
>
> A: Thanks for the suggestion, we will add more descriptions and captions for Figure 1,2,3.

---

> > ### Comment · Reviewer_LAyY · 2021-08-25
> > **Thank you for your rebuttal.**
> >
> > However, I still have some more questions:
> >
> > Q1: Thank you to the authors for answering how. But please clarify how to get the fake instances. To some degree it is quite important due to the listing reasons.
> >
> > Q3 and Q4: I believe that the authors should provide a more detailed plan on improving the writing and especially the figures since they are not minor issues in this submission (also pointed out by other reviewers). I think such writing and figures are not qualified for a NeurIPS paper.

---

> > > ### Author Response · Authors · 2021-08-26
> > > **Response to reviewer LAyY. Thanks for the reply.**
> > >
> > > Thanks for the reviewer's reply, we will keep on answering questions and revising our manuscript.
> > >
> > > --------------------
> > >
> > > **Q1: Please clarify how to get the fake instances. To some degree it is quite important due to the listing reasons.**
> > >
> > > A: Thanks for the question, we are pleased to provide more details. The category of each fake instance is drawn from a distribution, where the probability for each class is proportional to their frequency in the training set. At the same time, the location for each fake instance is generated by sampling a center position and a bounding box size. Empirically, we assume that widths and heights of bounding boxes of each class obey the Gaussian distribution. We calculate the statistical mean and variance according to the training set, the fake bounding box sizes are drawn from this Gaussian distribution. Then, center positions are sampled inside the image uniformly. Lastly, we clip bounding boxes that grow outside the image.
> > >
> > >
> > > **Q3 and Q4: I believe that the authors should provide a more detailed plan on improving the writing and especially the figures since they are not minor issues in this submission (also pointed out by other reviewers).**
> > >
> > > A: Thanks for taking care of the writing and figures, we also thank suggestions by all reviewers. We have made the following modifications, and we will continue to check and revise our manuscript.
> > >
> > >
> > > **Figure 1:**
> > >
> > > 1) We revise the captions and add sub-captions for each sub-figure. We also add more details for detection KD.
> > >
> > >
> > > 2) We add descriptions in the figure to explain the meaning of each component, e.g., teacher / student, proposal predictions, feature maps, discriminative areas, knowledge conditioned on each instance.
> > >
> > > 3) We unify the style of arrows and add new arrows to denote backpropagation, we also add a legend to describe the meaning of arrows. We hope this could explain how the distillation loss updates the student.
> > >
> > > 4) We add the loss function to the figure for a clear explanation, e.g., KL loss in logits-level, mimicking loss in proposal levels, feature-based loss, and our attention weighted loss.
> > >
> > > 5) We revise the color to make texts more discriminative. We revise components in the figure to make them more consistent from left to right.
> > >
> > > > Captions: Comparison of different methods for KD. (a) Conventional KD for classification transfers image-level predictions. (b) Some recent detection KD methods propose to sample regions for distillation, e.g., based on proposal predictions, label assignments, or activations. (c) The proposed method explicitly learns what knowledge should be distilled, it learns to find the knowledge that correlates with each instance by query-based attention.
> > >
> > > > (a)(b)(c) denotes the left, middle, and right columns in the current figure.
> > >
> > > **Figure 2:**
> > >
> > > 1) We revise the captions of this figure to add more descriptions.
> > >
> > > 2) We add projection layers in the figure and we modify rectangle areas. We also add descriptions to explain the meaning of components, we highlight modules for the decoding procedure and modules for distillation.
> > >
> > > 3) We revise the style of arrows and add new arrows to denote backpropagation, similar to the modification in Figure 1. We hope this could illustrate how the distillation loss updates the student and how the auxiliary task updates the decoding modules.
> > >
> > > > Captions: An overview of our approach. First, we compute instance information as a condition. We design a decoding procedure to learn instance-aware attention, it computes the correlation between teacher features and each given condition. Then, an auxiliary task is introduced to update parameters w.r.t. to the decoding procedure. Finally, we propose the feature-level distillation loss weighted by the instance-aware attention to update the student.
> > >
> > > **Figure 3:**
> > >
> > > 1) We revise the captions and description of the figure. We add sub-captions for each sub-figure.
> > >
> > > 2) We add a brief loss formulation in the figure for the identification task and localization task. We hope this could explain how we optimize the auxiliary task.
> > >
> > > 3) We revise the color and instance annotations, to make all components more discriminative.
> > >
> > > > Captions: The illustration of the auxiliary task. (a) The instance encoding function encodes a category, a position, and a scale to a vector, it is then projected to a query feature, which encodes a condition. (b) The identification task learns to identify the existence of a conditioned instance. (c) The localization task learns to predict a precise bounding box given an uncertain position provided by a condition.
> > >
> > > **Writing:**
> > >
> > > We have revised the concerns mentioned by reviewers, thanks again. We are checking and revising the manuscript elaborately to resolve any potential concerns.
> > >
> > > Our main modifications are as follows:
> > >
> > > 1) In general, we revise the phrases and add details for clear explanations, we also fix typos and rephrase less rigorous expressions (**Ccfq:Q5**, **LAyY:Q3**). We highlight the novelty of our method: how to explicitly retrieve informative knowledge for distillation in object detection.
> > >
> > > 2) For related work, we revise the description and add more details to compare with recent methods. We discuss more backgrounds of KD, that explains the improvement of our method (**WeoR:Q1/Q2/Q4**). We discuss the supervised training setting of KD that our method does not introduce extra data or labels (**WeoR:Q2**).
> > >
> > > 3) For methods, we add a more detailed explanation, we add a short discussion of fake instances and explain how to generate them (**LAyY:Q1**).
> > >
> > > 4) For experiments, we revise the description to emphasize our fair experiment settings (**Ccfq:Q1**,**WeoR:Q1/Q2/Q4**). We list the number of parameters and latencies in Table 2, which shows that our method does not introduce extra computations or parameters during inference (**zCpx:Q2**). Suggested experiments are added to the Appendix, all experiments show the good generalization ability of our method (**zCpx:Q1**, **Ccfq:Q1/Q2**, **LAyY:Q2**, **WeoR:Q5**).
> > >
> > > 5) For discussions, we discuss the training memory consumption, provide a solution for memory-efficient training (**Ccfq:Q3**). We add extra visualizations to the Appendix to illustrate more cases and explain that degenerated cases for heads are not usual (**Ccfq:Q4**).
> > >
> > > -----------------------
> > >
> > > We hope our reply can address the reviewer's concern.

---

> > > > ### Comment · Reviewer_LAyY · 2021-08-30
> > > > **Post-rebuttal**
> > > >
> > > > Thank you for the rebuttal. I will keep my score of 6. Hope the authors revise their paper accordingly and add the details of fake instances generation accordingly.

---

### Official Review · Reviewer_Ccfq · 2021-07-16

**Rating:** 7
**Confidence:** 4

**Summary:**

The paper proposes a target-oriented solution towards knowledge distillation for object detection. Unlike prior works which focus on distilling the global representations or foreground-region representations, the paper aims to find relevant regions for distillation based on the instance-conditional information. The auxiliary task optimization helps in finding the related knowledge. The paper performs extensive experiments on MS-COCO to show the efficacy of their method.

**Limitations And Societal Impact:**

The authors have addressed the limitations for their method in the discussion section.

1. Fail cases in the form of learning some degenerate heads.

2. Learning from a teacher can be costly.

3. Student might inherit some bias from the teacher.

However, Points 2 and 3 are the limitations of the Knowledge Distillation setup in general, not particular to this method.

**Main Review:**

- The idea of instance-aware knowledge distillation is novel.

- The paper achieves respectable performance improvement over the previous approaches even beats the teacher in
some cases.

- In Table 1, the backbone of the teacher network used is Resnet-101. For a fair comparison with [47],  the backbone
that should be used is Cascade Mask RCNN with ResNetXt101. [47] obtains 41.5% mAP on Resnet-50 Faster-RCNN student in this setting. Similar setting for RetinaNet would be appreciated. The method beats Wang et al. [42] with a good margin though.

- Although the paper argues that the cost associated with the additional training is negligible, the concatenation of the feature maps across different scales (L) seems to add more parameters for training.

- The method seems to depend on FPN. Is that the case ? This will limit the usage of the method significantly.

- In Figure 4, the learned attention over the heads is degenerate for some heads (v). Better techniques for learning
attention over each head should be explored.

- Correction: Line no. 229, I think it should be “a very recent work with strong performance from Zhang et al. [47]”.

**Time Spent Reviewing:**

7

---

> ### Author Response · Authors · 2021-08-10
> **Response to reviewer Ccfq. Thanks for the feedback!**
>
> We thank the reviewer for the careful review and valuable comments. We appreciate that the reviewer finds our method novel, our experiments extensive, and our performance respectful. We hope our answer can address the reviewer’s concerns.
>
> ---------------
>
> **Q1: Comparison with Zhang et al. [47].**
>
> A: For a fair comparison in Table 1, we conduct all experiments on the same baseline, where we adopt the same training schedules and teacher-student pairs for all methods.
>
> To compare with the reported performance of  [47], we implement our method on a similar setting, i.e., with longer schedules (2x) and single-scale training. Our baseline has lower performance compared with [47], due to the difference in the codebase. We also adopt relatively weaker teachers, as teachers of [47] are not available in our codebase. Our method still outperforms [47] in this setting, especially for RetinaNet.
>
> |                   | Zhang et al. [47]  | ICD (Ours)    | Zhang et al. [47]  | ICD (Ours)   |
> |-------------------|--------------------|---------------|--------------------|--------------|
> | Student Detector  | Faster R-CNN       | Faster R-CNN  | RetinatNet         | RetinaNet    |
> | Teacher AP        | 47.3               | 45.1          | 41.0               | 40.4         |
> | Student AP        | 38.4               | 38.2          | 37.4               | 36.2         |
> | + Ours AP         | 41.5 (+3.1)        | 41.6 (+3.4)   | 39.6 (+2.2)        | 40.5 (+4.3)  |
>
> **Q2: The reliance of FPN.**
>
> A: Our method does not rely on FPN. Specifically, on Faster R-CNN C4 and Faster R-CNN DC5 that do not use FPN, our method still achieves prominent improvements, as shown below on MS-COCO.
>
> |             | Faster R-CNN C4  | Faster R-CNN DC5  |
> |-------------|------------------|-------------------|
> | Teacher AP  | 41.1             | 40.6              |
> | Student AP  | 35.7             | 37.3              |
> | + Ours AP   | 38.5 (+2.8)      | 39.3 (+2.0)       |
>
> **Q3: Training cost.**
>
> A: Our method consumes a small portion of time (3%) and a little extra memory (~200MB), we believe this training cost is affordable under most situations.
>
> 1) The decoder training and distillation only takes a small portion of time (3%) in the full training cycle, as reported in Figure 5.
>
> 2) The decoding modules introduce a few parameters (2MB) and consume a little extra memory (~200MB) during training. Note that the training of the auxiliary task is independent of the student, by updating the decoder and the student in separate iterations (with the same data), only features for distillation and related gradients consume extra memory. For example, training a regular student alone takes about 4.4GB max memory for RetinaNet, training the decoding procedure alone takes 2.4GB, the max consumption for ICD is only about 4.6GB.
>
>
> **Q4: Degenerated heads and better learning techniques.**
>
> A: We do observe degenerated heads in some cases, yet the final performance has not been much influenced. We are trying some new techniques, such as regularizations and other positional embeddings, we will report them as variations if possible. Yet improving attention learning is not the main purpose of this paper.
>
>
> **Q5: Correction on Line no. 229.**
>
> A: Thanks. We will fix it.

---

> > ### Comment · Reviewer_Ccfq · 2021-09-02
> > **After rebuttal**
> >
> > I thank the authors for their detailed response and it answers most of my original queries. I have also gone through the other reviewers' feedbacks and authors' responses to them. Based on all these, I find the paper good enough to be accepted, and updated my rating to 7. Hope the authors include all the suggested changes/additions in the revised version of the paper.

---

### Official Review · Reviewer_zCpx · 2021-07-20

**Rating:** 7
**Confidence:** 5

**Summary:**

This paper propose a novel method for knowledge distillation on object detection by measuring the correlation between each object and human observed instances. They have introduced a conditional modeling paradigm to find the crucial regions for distillation and apply the auxiliary tasks to supervise its training. Good experiment results have been achieved on COCO.

**Limitations And Societal Impact:**

1. All the experiments are conducted on COCO, it will be better with experiments on more datasets such as Cityscpaes.
2. In table 2, it will be better to report the inference time and parameters of each model.
3. Figure 1 "Previous studies for object detection distill discriminate regions by sampling proposals or center areas". This is not correct. There have been some methods which find the regions without proposals such as [47].
4. Figure 2 needs to be explained better.

**Main Review:**

Originality. The ideas that applying KD on object detection and find the important regions for KD are not new. But the ideas that applying transformer decoder and conditional computation to find the KD regions are very interesting and novel.

Quality. This paper has sufficient experiments and ablations to show the effectiveness of their methods. Although there is no theoretical proof about their methods, the visualization results provided in the paper show that they can find better KD regions in object detection. I think it's a new technical sound in this domain.

Clarity. This paper is easy to follow but some details need to be improved.

Significance. They have outperform the existing object detection KD methods by a clear margin in Table.1. And good results have been achieved on the state-of-the-art detectors such as SOLOv2 and CondInst and efficient detectors such as EfficientNet-B0.



**Time Spent Reviewing:**

4

---

> ### Author Response · Authors · 2021-08-10
> **Response to reviewer zCpx. Thanks for the feedback!**
>
> We thank the reviewer for the careful review and valuable comments. We appreciate that the reviewer finds our method interesting and novel, and recognizes the good performance and possibility of our work. We hope our answer can address the reviewer’s concerns.
>
> --------------
> **Q1: Experiments on more datasets like Cityscapes.**
>
> A: On more datasets like Cityscapes and VOC, our method still has considerable improvement. The results are reported in the Supplementary material, Appendix B, Table 2 and Table 3. For instance, our method leads to 3.6 Mask AP gains on Mask R-CNN on Cityscapes.
>
> | Method   | Backbone      | Mask R-CNN (AP)  | Mask R-CNN (AP50)  |
> |----------|---------------|------------------|--------------------|
> | Teacher  | ResNet-50     | 33.5             | 60.8               |
> | Student  | MobileNet V2  | 28.8             | 55.2               |
> | + Ours   | MobileNet V2  | 32.4             | 60.1               |
>
> Thanks for the suggestion, we will emphasize them in the main paper.
>
> **Q2: Report inference time and parameters in Table 2.**
>
> A: Thanks for the suggestion, we will add them to the table. Notably, our method does not cost extra time nor parameters during inference.
>
>
>
> **Q3: About Figure 1.**
>
> A: Thanks for the suggestion, we will revise our description.
>
>
>
> **Q4: About Figure 2.**
>
> A: Thanks for the suggestion, we will add more descriptions for Figure 2.

---

> > ### Comment · Reviewer_zCpx · 2021-08-28
> > **Thanks for author response**
> >
> > Thanks for author response. Most of my concerns have been solved and I think this paper is good enough to be accepted.

---

### Decision · Program_Chairs · 2021-09-27

**Decision:**

Accept (Poster)

**Comment:**

Although the paper originally received slightly mixed ratings, with one reviewer recommending rejection and 3 acceptance, the authors' feedback eventually convinced the most negative reviewer to update their score. Altogether, the reviewers acknowledge the novelty of the proposed method and that the empirical results are convincing. The authors' feedback also provided additional results and clarifications, which we strongly encourage the authors to incorporate in the final version.